# Dewatering-Induced Stratified Settlement around Deep Excavation: Physical Model Study

Xiaotian Liu [1,2,3], Jianxiu Wang [1,3,*], Tianliang Yang [4,5], Lujun Wang [6], Na Xu [1], Yanxia Long [1] and Xinlei Huang [4,5]

1 College of Civil Engineering, Tongji University, Shanghai 200092, China
2 China Construction Eighth Engineering Division Corp., Ltd., Shanghai 200112, China
3 Key Laboratory of Geotechnical and Underground Engineering of Ministry of Education, Department of Geotechnical Engineering, Tongji University, Shanghai 200092, China
4 Shanghai Institute of Geological Survey, Shanghai 200072, China
5 Key Laboratory of Land Subsidence Monitoring and Prevention of Ministry of Natural Resources, Shanghai 200072, China
6 Center for Hypergravity Experimental and Interdisciplinary Research, Key Laboratory of Soft Soils and Geoenvironmental Engineering of Ministry of Education, College of Civil Engineering and Architecture, Zhejiang University, Hangzhou 310058, China
* Correspondence: wangjianxiu@tongji.edu.cn; Tel.: +86-139-1618-5056 or +86-21-65983036; Fax: +86-21-65985210

**Featured Application: This research can be widely applied to the risk control of groundwater and environmental set-tlement during deep excavation.**

**Abstract:** The multi-aquifer and multi-aquitard system (MAMA) is a typical geological structure in deltas. Thus, the risks and challenges to settlement control and environmental protection are increased when demand for underground space extends to deeper strata. In this study, dewatering-induced stratified settlement in MAMA is divided into three stages according to whether the overlying aquitard is coupled with groundwater seepage. Subsequently, large physical model tests were carried out. Seepage and compression in the overlying strata come after the compression in the confined aquifer and the coordinated deformation in the overlying strata. The soil is compressed under the seepage drive within the hydraulic gradient range, while the soil above it is still affected by coordinated deformation and shows expansion. Dewatering-induced uneven settlement will cause damage to existing foundations and underground structures. Large-scale and uninterrupted excavation and dewatering are the main reasons for the continuous development of land subsidence. Although artificial groundwater recharging can reduce the settlement of the existing building, underground structure, and surrounding strata, a reasonable space arrangement is needed.

**Keywords:** multi-aquifer and multi-aquitard system; stratified settlement; foundation pit dewatering; physical model test; coordinated deformation

## 1. Introduction

Land subsidence caused by fluid withdrawal has a long history and is widely distributed all over the world [1–4]. It is a widespread and common disaster because of elevation loss, earth fissures, and structural damage, which increase flood susceptibility accompanied by groundwater depletion risk [5,6]. This field entered researchers' vision formally when Terzaghi [7] proposed the principle of "effective intergranular stress", which established the relationship between underground fluid seepage and stratum consolidation. During underground space exploitation, secondary disasters caused by groundwater dewatering constantly emerge, such as foundation settlements of adjacent structures [8], water leakage of adjacent tunnels [9], retention of structure displacement of deep excavation [10,11], loss of groundwater resources [12], ground settlement [13], and earth

fissures [14]. Large-scale and uninterrupted deep excavation and dewatering have been recognized as the main causes of urban land subsidence [15,16].

Wang et al. [17] first discovered the coordinated deformation phenomenon of strata during the settlement monitoring of the deep excavation dewatering in Yishan Road Station of Shanghai Metro Line 9 and reproduced it in physical model tests [18]. Loáiciga [19] and Zhang et al. [20] constructed 3D mathematical models to describe this phenomenon in multi-layer strata.

As a kind of intuitive and generalized method, the physical model test can reproduce the real environment by similarity theory [21] in a laboratory to eliminate the interference factors and analyze the stress and strain law of the research object under specific loading and boundary conditions. Among the physical model tests, the 1D physical model test, which is mainly used to study stress and strain response law in vertical soil under a complete lateral confinement condition, is the simplest. By analyzing the deformation rule of the combined strata under the condition of pumping water in the 1D physical model, deformation lags behind groundwater variation, and vertical uneven settlement in strata was observed [22,23]. After repeated groundwater-level fluctuations, the compression of pores and the collapse of a card-house structure prolonged the consolidation time and tended to stabilize plastic and creep deformation; moreover, the overall performance is elastic deformation [24,25]. Restricted by the boundary limited in the 1D model, scholars began to explore the deformation characteristics of strata in space and time by using a 2D or 3D physical model test. Wang et al. [18,26] reproduced the phenomenon of coordinated deformation and considered that the expansion of overlying strata is caused by coordination deformation, while compression is caused by overflow and consolidation. According to Cui et al. [27], groundwater withdrawal changed the subsidence distribution, which developed at a building load period during the dewatering of the friction force from piles. These piles reduced the subsidence of buildings and the surrounding area yet increased the differential subsidence based on the physical model test of the variation in the groundwater level with the high-rise building group.

For deep excavation dewatering, the deformation of strata caused by the decrease in pore water pressure and the increase in effective stress is more complicated. Moreover, this deformation entails complex rules with the development of the dewatering time because of the complex urban environment and the high settlement control requirements. Thus, to distinguish the deformation mechanism under each factor, dewatering-induced stratified settlement in the multi-aquifer and multi-aquitard system (MAMA) can be divided into three stages according to whether the overlying aquitard coupled with groundwater seepage: compression in a confined aquifer, coordinated deformation in overlying strata, and seepage and compression in overlying strata. The first two stages mainly occur in the early phase of dewatering or when the overlying aquitard's hydraulic conductivity is very low. The last stage mainly occurs in the later period of dewatering or when the overlying aquitard's hydraulic conductivity is larger. Furthermore, the overlying strata exhibit progressive vertical groundwater seepage under the action of the hydraulic gradient.

## 2. Background

Shanghai, which is located in the southeast of the Yangtze River Delta, has a relatively flat terrain with a ground elevation ranging from 2.2 to 4.8 m, except for scattered mounds with an elevation that does not exceed 100.0 m in the southwest. The estuary of the Yangtze River is located in the north of Shanghai, wherein shoals and sandbanks are formed under the combined action of rivers and sea tides, which then developed into sand islands. The Quaternary strata in Shanghai are formed under the action of river water and sea tide, and the deposition thickness is approximately 200.0–350.0 m. The area has an alternating strata structure of sand, silty, silty clay, clay, and muddy clay vertically, including 16 engineering geological strata (Table 1). According to the hydrological characteristics of the strata, the strata can be divided into one phreatic aquifer, one micro confined aquifer, five confined

aquifers, and six aquitards [28], which are collectively called the MAMA by [29] or the multi-aquifer aquitard systems (MAAS) by [30]. This system in Shanghai is not complete or independent but is a part of the Yangtze River Delta groundwater system.

**Table 1.** Engineering geological characteristics in Shanghai.

| Strata Calendar | | Genetic Category | No. | Geological Characteristics | Hydrogeology and Engineering Geology Type |
|---|---|---|---|---|---|
| Holocene ($Q_4$) | $Q_4^4$ | Estuarine facies, littoral facies | ① | Miscellaneous fill | Phreatic aquifer (PAq) |
| | $Q_4^3$ | Estuarine facies, littoral facies | ②1 | Clay | |
| | | | ②2 | Clay, mucky clay | |
| | | | ②3 | Silty soil, silty sand | |
| | $Q_4^2$ | Littoral facies, neritic facies | ③ | Mucky silty clay | First soft soil (SSo–I) |
| | | | ④ | Mucky clay | |
| | $Q_4^1$ | Littoral facies, swamp facies | ⑤1 | Clay | Second soft soil (SSo–II) |
| | | | ⑤2 | Sandy silt, silty sand | Micro confined aquifer (MCAq) |
| | | Drowned river valley | ⑤3 | Silty clay, clayey silt | Second soft soil (SSo–II) |
| | | | ⑤4 | Silty clay, clayey silt | |
| | | Lacustrine facies, swamp facies | ⑥ | Clay | First hard soil (HSo–I) |
| Upper Pleistocene ($Q_3$) | $Q_3^2$ | Estuarine facies, littoral facies | ⑦1 | Silty sand | First confined aquifer (CAq–I) |
| | | | ⑦2 | Silty fine sand | |
| | | Littoral facies, neritic facies | ⑧1 | Clay | Second hard soil (HSo–II) |
| | | | ⑧2 | Silty clay, silty sand | |
| | $Q_3^1$ | Littoral facies, estuarine facies | ⑨1 | Silty sand, fine sand | Second confined aquifer (CAq–II) |
| | | | ⑨2 | Medium–coarse sand, fine sand | |
| Middle Pleistocene ($Q_2$) | $Q_2^2$ | Estuarine facies, lacustrine facies | ⑩ | Clay | Third hard soil (HSo–III) |
| | $Q_2^1$ | Estuarine facies, littoral facies | ⑪ | Silty fine sand, medium–coarse sand, gravel | Third confined aquifer (CAq–III) |
| Lower Pleistocene ($Q_1$) | $Q_1^3$ | Lacustrine facies | ⑫ | Clay | Fourth hard soil (HSo–IV) |
| | $Q_1^2$ | Fluvial facies | ⑬ | Medium–coarse sand, gravel | Fourth confined aquifer (CAq–IV) |
| | | Lacustrine facies | ⑭ | Clay | Fifth hard soil (HSo–V) |
| | $Q_1^1$ | Fluvial facies | ⑮ | Medium–fine sand | Fifth confined aquifer (CAq–V) |
| | | Lacustrine facies | ⑯ | Clay, gravel | Sixth hard soil (HSo–VI) |

At present, human engineering construction mainly focuses on layers ① to ⑨, and only a few groundwater mining, artificial recharging, and geothermal resource developments come into contact with deep aquifers (Figure 1). Unlike the continuous deposition of layers ⑤1 and ⑥ of clay in a normal sedimentary area, part of layers ⑤1, ⑥, and ⑦ were eroded by the paleochannel of the Suzhou River, and redeposited layers ⑤2, ⑤3, and ⑤4. Considering that the hydraulic conductivity of these three layers is better than those of layers ⑤1 and ⑥, the groundwater level change in the first confined aquifer (layer ⑦) is transmitted to the overlying strata more easily. Hence, according to the distribution of paleochannel deposition and the contact relationship between layers ⑦ (first confined aquifer) and ⑨ (second confined aquifer), the strata in the coastal plain area of central Shanghai can be divided into four types: I. normal deposition area where layers ⑦ and ⑨ are connected; II. paleochannel deposition area where layers ⑦ and ⑨ are connected; III. normal deposition area where layers ⑦ and ⑨ are disconnected; and IV. paleochannel deposition area where layers c and ⑨ are disconnected [29].

In this study, the deepest underground excavations under construction for the Deep Drainage and Storage Pipeline Project under Suzhou Creek in Shanghai are selected as the engineering background. This project aims to solve the problem of waterlogging in Shanghai. It consists of nine deep excavations, a 15.3 km long main drainage tunnel, and other supporting facilities. The average buried depth of the tunnel is 60 m. Deep excavation in West Yunling Road is selected as the prototype of the physical model, and the complex environment around the deep excavation is simulated by hypothetical tunnels and buildings. To examine the law of stratified settlement in MAMA during deep excavation dewatering in Shanghai under the most unfavorable MAMA conditions, a typical area under the paleochannel deposition area where layers ⑦ and ⑨ are connected is selected as the geological background.

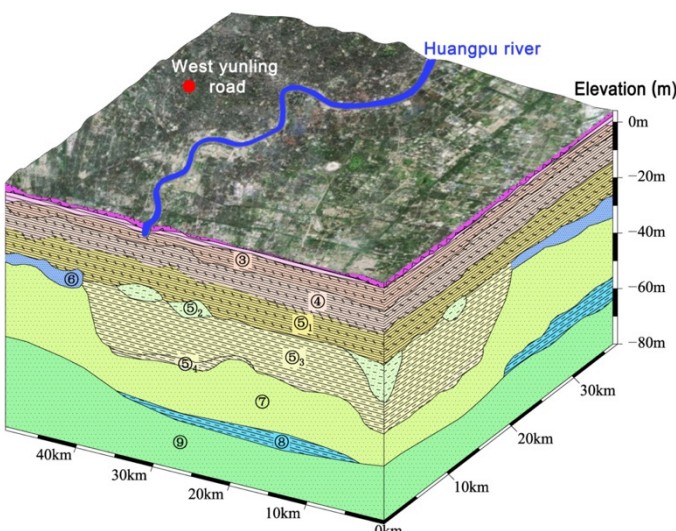

**Figure 1.** Typical geological structure of shallow strata in Shanghai (satellite images cited from Google Maps).

## 3. Materials and Methods

### 3.1. Artificial Assumption Boundary

A hydraulic connection between a confined aquifer and an adjacent aquitard is observed in the MAMA system for general soil layers in Shanghai. During dewatering in the confined aquifer, the pore water pressure of the overlying strata changes correspondingly. In the dewatering process, the pore water pressure in the confined aquifer decreases and breaks the original stress balance. According to the principle of effective stress, the decrease in pore water pressure in the confined aquifer is equivalent to the application of negative pore water pressure inside the MAMA system, which is an axisymmetric funnel with the center of the pumping well or deep excavation. Therefore, to simulate this physical and mechanical process in the physical model tests, the similarity ratio (prototype/physical model) of the basic physical parameter force ($F$), length ($L$), and time ($T$) are defined as $\alpha_L = 150{:}1$, $\alpha_F = 150^3{:}1$ , and $\alpha_T = 15{:}1$ , respectively. According to the *FLT* dimensional analysis method, the similarity ratio of other physical parameters in the MAMA deformation system is shown in Table 2.

**Table 2.** Dimensional analysis of main parameters in physical model tests.

| Parameters | Symbol | Dimensional Analysis | Similarity Ratio (Prototype/Physical Model) |
|---|---|---|---|
| Stratum's thickness | $H$ | $H = L$ | 150:1 |
| Weight | $\gamma$ | $\gamma = \rho g = \frac{FT^2}{L^4}\frac{L}{T^2} = \frac{F}{L^3}$ | 1:1 |
| Young's modulus | $E_s$ | $E_s = \frac{F}{L^2}$ | 150:1 |
| Hydraulic conductivity | $K$ | $K = \frac{g}{v}k = \frac{L}{T^2}\frac{T}{L^2}L^2 = \frac{L}{T}$ | 10:1 |
| Groundwater level drawdown | $\Delta h$ | $\Delta h = L$ | 150:1 |
| Pore water pressure | $u$ | $u = \frac{F}{L^2}$ | 150:1 |
| Settlement | $\Delta H$ | $\Delta H = L$ | 150:1 |

Quaternary strata in middle and shallow portions can be simplified into five layers, namely, ① + ② of artificial filled soil, ③ + ④ of mucky clay, ⑤ of silty clay, ⑦ of fine sand, and ⑨ of medium–coarse sand. According to long-term hydrological observation data of Shanghai, the depth of the natural groundwater level of the first and second confined aquifers is 3.0–11.0 m and 5.0–12.0 m, respectively. Given that two layers (e.g., layers ⑦ and ⑨) of confined water were connected in this test, the natural depth of the groundwater level in the prototype and in the physical model is set as 3.0 m and 20.0 mm, respectively. To meet the similar requirements of the physical model test, natural clay (taken from layer ④)

is selected as the base material after drying and crushing by adding different proportions of water, fine sand, and sponge to meet the similarity ratio of weight $\gamma$, Young's modulus $E_s$, and hydraulic conductivity $K$ (Table 3).

**Table 3.** Parameters of similar materials.

| Stratum | Materials | Thickness (mm) | Young's Modulus, $E_s$ (MPa) | Hydraulic Conductivity, $K$ (cm/s) | Weight, $\gamma$ (kN/m³) |
|---------|-----------|----------------|------------------------------|-------------------------------------|--------------------------|
| ① + ② | Counterweight by medium–coarse sand | 40.00 | – | – | 18.30 |
| ③ + ④ | Mixture (weight ratio 30.0:2.3:0.6, clay–water–sponge) | 140.00 | $E_{s(0.5-1.0 kPa)} = 0.015$ | $4.18 \times 10^{-7}$ | 17.10 |
| ⑤ | Mixture (weight ratio 15.0:15.0:1.5:0.6, clay–fine sand–water–sponge) | 130.00 | $E_{s(2.5-5.0 kPa)} = 0.038$ | $9.47 \times 10^{-7}$ | 18.90 |
| ⑦ | Fine sand | 250.00 | $E_{s(6.3-12.5 kPa)} = 1.984$ | $4.31 \times 10^{-4}$ | 17.80 |
| ⑨ | Medium–coarse sand | 500.00 | $E_{s(12.5-25.0 kPa)} = 6.944$ | $1.16 \times 10^{-3}$ | 18.30 |

### 3.2. Artificial Assumption Boundary

In the prototype, the horizontal influence range of dewatering-induced groundwater level changes can reach several kilometers [31,32], which cannot reappear in a physical model test with limited size. Therefore, the variable hydraulic boundary is adopted in the model to eliminate the lateral boundary effect, and the groundwater level in the lateral boundary can be determined by numerical calculation. During the test, the groundwater levels in the lateral boundary and inside the deep excavation are reduced simultaneously, and a funneled groundwater level is formed around the deep excavation. This method can reduce the constraint effect of groundwater boundary conditions in the lateral boundary effectively and simulate the true funneled groundwater level to the greatest extent.

The size of the model test area is 3.0 m long, 3.0 m wide, and 1.5 m high. Two layers of variable hydraulic boundary are set on the inside wall and buried in two layers of confined aquifers (layers⑦and⑨). Two circulating water supply systems are set outside to control the groundwater levels of two variable hydraulic boundaries separately. As shown in Figure 2, the hydraulic boundary, including four layers of octagonal closed PVC pipes with holes, with two layers connected to a circulating water supply system as a group, can evenly provide a stable groundwater level to the surrounding boundary of the confined aquifer.

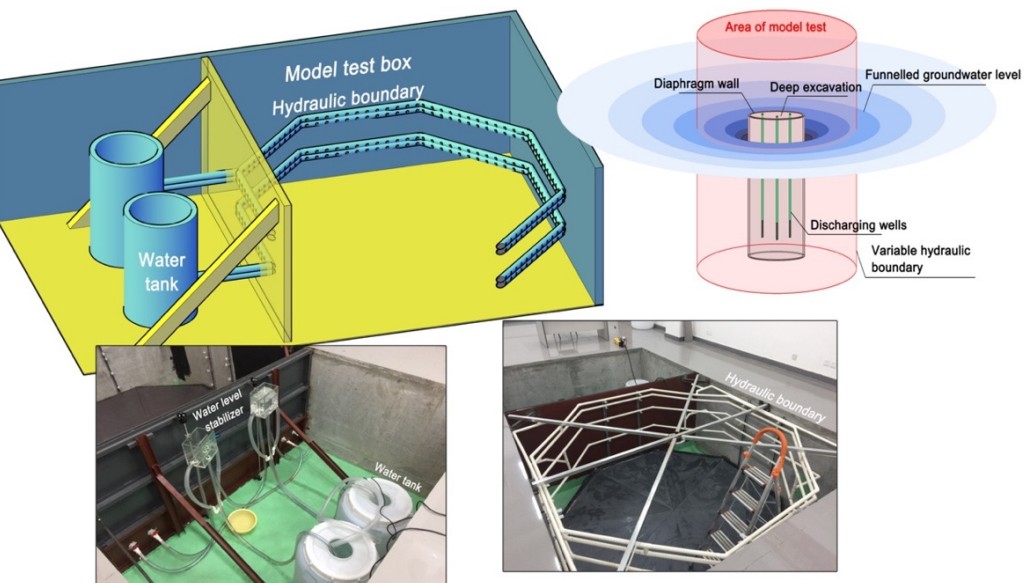

**Figure 2.** Structure of model test box with artificial assumption boundary.



### 3.3. Underground Structure Model

As the length similarity ratio $\alpha_L$ = 150:1, the size of the deep excavation model can be calculated and shown in Table 4. The diaphragm wall of the deep excavation model is simulated by the PMMA tube, and the outer wall is rough polished to improve its waterproofing ability and friction. The planar multi-point uniform distribution method is used to simulate dewatering inside the deep excavation. The discharging wells are perforated steel pipes covered with 300-mesh gauze. In addition, the distribution under the collection device is uniform. Thus, groundwater can be centrally discharged with a constant flow by a peristaltic pump. Linear multi-point uniform distribution method is used to simulate groundwater recharging (spacing is 200.0 mm). Moreover, the recharging wells are set parallel to the tunnel model near the deep excavation model, and constant water pressure is used during the test.

**Table 4.** Similar parameters of underground structure model.

| Category | Deep Excavation Model | | | | Discharge/Recharge Well | |
| --- | --- | --- | --- | --- | --- | --- |
| | Inner Diameter | Thickness | Excavation Depth | Buried Depth | Well Screen Length | Outer Diameter with Sand Filter |
| Prototype (m) | 30.00 | 1.50 | 59.60 | 105.00 | 30.00 | 1.20 |
| Physical model (mm) | 200.00 | 10.00 | 400.00 | 700.00 | 200.00 | 8.00 |

To analyze the dewatering-induced environmental impact around the deep excavation, the building models, tunnel model, and monitoring points for stratified settlement, groundwater level, and earth pressure are set around the deep excavation model (Figure 3). Three building models with end-bearing pile foundations are set beside the deep excavation model, and the actual piles are equivalently replaced with eight PMMA (polymethyl methacrylate) piles that are evenly distributed on the plane (spacing is 50.0 mm), with a diameter of 14.0 mm and a length of 360.0 mm. A $100.0 \times 200.0 \times 1.5$ mm steel plate covers the PMMA piles to simulate the foundation slab, and a 25.0 kg weight is placed on it to simulate the building load.

A single-track tunnel model is set beside the deep excavation model. It is simplified as a plastic-coated metal hose with an outer diameter of 51.0 mm in the physical model, and steel joints are set in the tunnel model every 300.0 mm to connect the settlement poles and the earth pressure sensors. The tunnel model was buried in layer ③ + ④ of mucky clay, and two $150.0 \times 150.0 \times 400.0$ mm PMMA boxes are set at both ends to simulate the underground structures of the subway station. Compared with the prototype tunnel, the tunnel model has lower stiffness and larger deformation range, which are more conducive to the analysis of the deformation law.

To avoid the influence of the sensor on soil deformation, small-sized sensors were selected. The size of the pore water pressure sensor is $\varphi$ 1.0 cm and 2.0 cm in thickness, and the size of the earth pressure sensor is $\varphi$ 2.5 cm and 0.5 cm in thickness. The settlement pole is made of a $\varphi$ 0.2 cm pole and two 1.5 cm $\times$ 1.5 cm plates at both ends, and a $\varphi$ 0.3 cm sleeve was placed on the outside of the 0.2 cm pole, so it can move freely along the vertical.

For the strata soil in the physical model, similar materials are filled from bottom to top in layers with a thickness of 50 mm. The underground structure model is pre-buried, and monitoring sensors are at an appropriate height. First, each layer of similar materials is compacted with a vibrating roller or a heavy hammer. Second, each layer is saturated with water from the bottom until the internal air is removed. To ensure that each layer of similar materials is uniform and meets the test design requirements, compacted samples of similar materials are collected for the density, water content, degree of consolidation, and hydraulic conductivity tests.

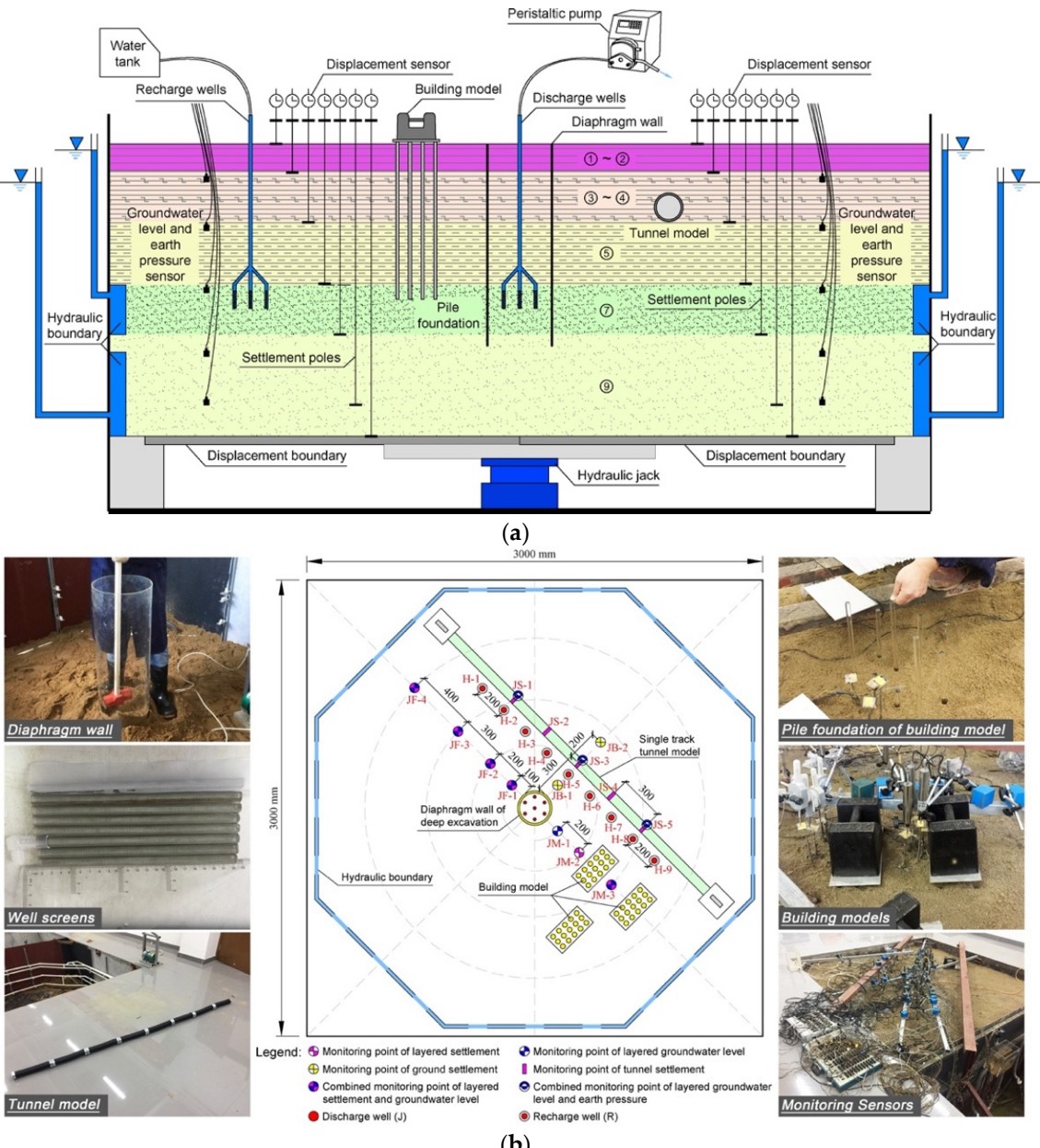

**Figure 3.** Layout of underground structure model and monitoring points: (**a**) Section view; (**b**) Plan view.

### 3.4. Test Conditions

A peristaltic pump is used to pump groundwater from dewatering wells to reduce the groundwater level inside the deep excavation. Meanwhile, the groundwater level in the variable hydraulic boundary has declined according to the pre-calculated results. This part of the test includes four stages:

- Graded dewatering: The gradual decrease in groundwater level is simulated during the layered excavation. Three groundwater level drawdowns are set inside the deep excavation at 150.0, 300.0, and 400.0 mm (22.5, 45.0, and 67.5 m in the prototype, respectively); the groundwater level drawdowns in the variable hydraulic boundary are set to 30.0, 60.0, and 80.0 mm in layer ⑨ (4.5, 9.0, and 12.0 m in the prototype, respectively) and 15.0, 30.0, and 40.0 mm in layer ⑦ (2.3, 4.5, and 6.0 mm in the prototype, respectively). The duration time under each groundwater level drawdown is set as 90.0 h, which is determined by the actual deep excavation process.

- Groundwater level recovery: In this stage, the deformation rebound of the strata during the recovery of groundwater level after deep excavation is analyzed. Pumping is stopped inside the deep excavation, the groundwater level in the variable hydraulic boundary recovers to its initial value, and the duration time is set as 170.0 h.
- Secondary dewatering: In this stage, the law of strata deformation caused by the secondary dewatering of the deep excavation under the condition of preconsolidation, which usually occurs when deep excavations are excavated by zones, is analyzed. The groundwater level drawdown inside the deep excavation is directly set as 450.0 mm, and the duration time is set as 190.0 h.
- Artificial recharging: In this stage, the control effect of the linear structure (tunnel) settlement under the action of artificial recharging in the process of deep excavation dewatering is analyzed. Water pressure in recharging wells is set as 1.0 kPa (100 mm initial groundwater level value), and the duration time is set as 110.0 h.

## 4. Results

Before testing, the strata in the physical model experienced 180 days of self-weight consolidation stage under the stable pore water pressure supply from the hydraulic boundary until the settlement of similar materials in each layer is stable.

### 4.1. Stratified Settlement of Strata in Open Spaces

Groundwater flows into the deep excavation driven by dewatering, thereby generating a funneled groundwater level around the deep excavation. However, given the existence of a diaphragm wall, the hydraulic connection of groundwater inside and outside the deep excavation is cut off within its depth range, and frictional water head loss is increased with the groundwater seepage path. In the graded dewatering stage, each time the groundwater level inside the deep excavation is low, the groundwater level changes in the confined aquifers reach stability after 1.5–3.5 days of dewatering, while reaching stability takes 35.0–40.0 days in aquitards (Figure 4). Combined with the strata deformation curve in Figure 5a, the strata deformation evidently falls behind the change in the groundwater level. In the confined aquifer, it reaches 80.0% of the total deformation on the 7th–9th day after the groundwater level has remained unchanged and achieves a stable state on the 20th–25th days. In aquitards, given the progressive vertical groundwater seepage and visco-elastic-plastic deformation, the stable state in the graded dewatering stage is reached at 35–40 days, and the groundwater level is recovered after 80 days. The time required for deformation stability and groundwater level is basically the same. Thus, the deformation of the clayey strata is mainly driven by groundwater seepage.

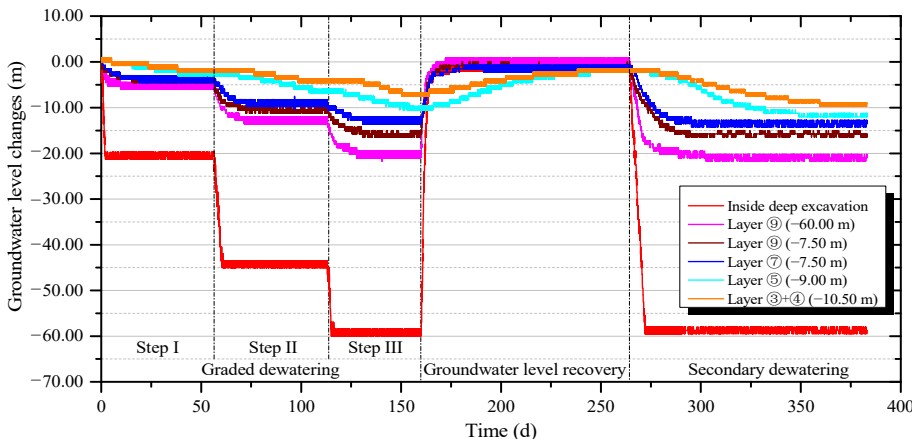

**Figure 4.** Groundwater level changes in strata at JF–1 (15.0 m away from deep excavation).

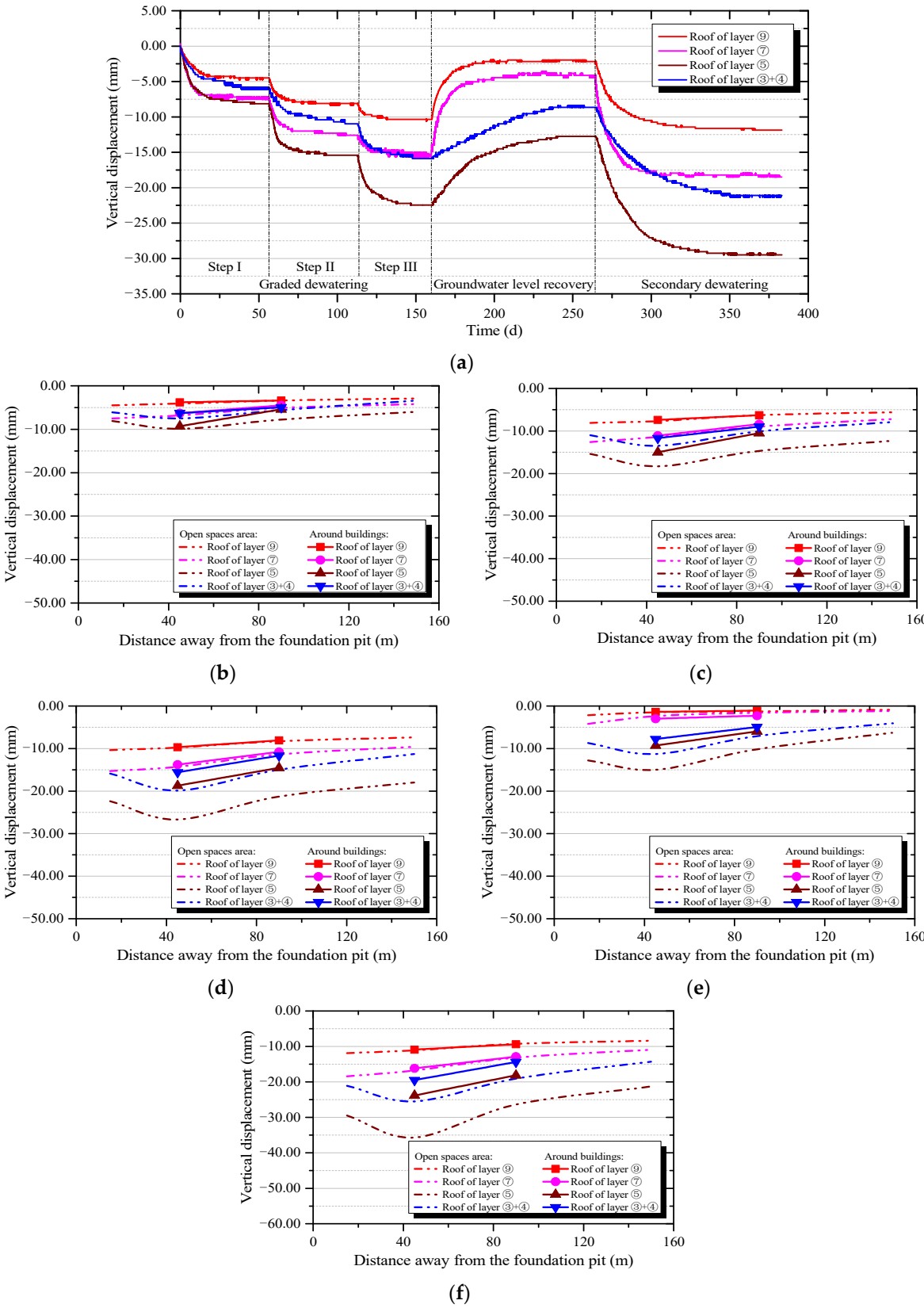

**Figure 5.** Stratified settlement in strata: (**a**) Stratified settlement with time in strata at JF–1 (15.0 m away from deep excavation); (**b**) Graded dewatering step I; (**c**) Graded dewatering step II; (**d**) Graded dewatering step III; (**e**) Groundwater level recovery; (**f**) Secondary dewatering.

As shown in Figure 5a, for the entire strata, layers ⑨, ⑦, and ⑤ exhibit compression deformation. Meanwhile, layer ③ + ④ displays expansion deformation. In addition, the maximum stratified settlement appeared at the top of layer ⑤, and the settlement of layer ③ + ④ is approximately 60.0–80.0% of layer ⑤. As layer ⑤ (silty clay) has certain hydraulic permeability ($K = 9.47 \times 10^{-6}$ cm/s in prototype), the vertical groundwater overflow will be driven by the hydraulic gradient from the underlying confined aquifer, and compression occurs inside the aquitard. For layer ③ + ④ (mucky clay), given its low hydraulic conductivity ($K = 4.18 \times 10^{-6}$ cm/s) and far vertical distance from the confined aquifer, the expansion caused by coordinated deformation accounts for the main part. Thus, compression caused by dewatering and the expansion caused by coordinated deformation occurs simultaneously in the confined aquifer. As shown in Figure 5b–f, the stratified settlement curve of the strata is similar to a spoon in the vertical section. The maximum value appears 45.0 m away from the deep excavation (measuring point JF–2), and the value decreases with the increase in distance. The frictional resistance of the diaphragm wall has an intense constraint effect on the deformation of the surrounding strata, and the settlement in the ground and clayey strata around the deep excavation is small (measuring point JF–1). When dewatering stops, the groundwater is replenished to the confined aquifer from the lateral boundary, and the groundwater level recovers gradually until the initial value is reached. The frictional resistance of the diaphragm wall still exists, thereby causing the rebound value of the strata deformation at 15.0 m away from the deep excavation (measuring point JF–1) to be smaller than those of other measuring points. Furthermore, the final settlement has rebounded by 70.0–90.0% (Figure 5e) because the deformation of sandy strata (layers ⑦ and ⑨) is mainly elastic. However, for the clayey strata (layers ③ + ④ and ⑤), the deformation presents visco-elastic-plastic characteristics, and the final settlement has rebounded by 40.0–65.0%. Compared with the graded dewatering stage, the final stratified settlement of secondary dewatering increases by 12.0–16.0% (layer ⑨), 14.0–21.0% (layer ⑦), 18.0–34.0% (layer ⑤), and 18.0–33.0% (layer ③ + ④). Therefore, repeated groundwater level changes will cause further settlement in the strata (Figure 5f).

### 4.2. Stratified Settlement of Strata around Buildings

When buildings are located around the deep excavation, the deformation of the clayey strata decreases by 22.0–32.0% in the graded dewatering stage (Figure 5b–d), and the stratified settlement difference between these two strata is not so noticeable because of the existence of the pile foundation's frictional resistance. Considering that layer ⑦ is the bearing layer of the buildings' pile foundation, the consolidation degree of the foundation layer is higher than that of the natural. Thus, the settlement under the buildings' load is approximately 5.0% less than that in open spaces (Figure 5b–d). In layer ⑨, the deformation is not affected due to its deep buried depth. In general, layers ⑨, ⑦, and ⑤ show compression deformation. Meanwhile, layer ③ + ④ shows expansion deformation, and the top of layer ⑤ shows the maximum stratified settlement. The existence of buildings has a certain inhibitory effect on the ground settlement.

When the groundwater level recovered, the pile foundation of the buildings undergoes an increase along with layer ⑦, which produces vertical upward frictional resistance and movement in the clayey strata, and the rebound ratio of the deformation is increased by 7.0% compared with that in open spaces (Figure 5e). Compared with the graded dewatering stage, secondary dewatering causes further settlement in the strata, and the final stratified settlement increases by 14.0–17.0% (layer ⑨), 17.0–20.0% (layer ⑦), 24.0–27.0% (layer ⑤), and 23.0–25.0% (layer ③ + ④) (Figure 5f). In this stage, the deformation in the clayey strata is affected by the frictional resistance of the pile foundation again, and the increase in the secondary settlement has reduced by 3.0–7.0% compared with that in open space. Given that the pile foundation of buildings, which is only 6.0 m, enters the bearing layer (layer ⑦), the piles' number and spacing are limited after generalization in the physical model. Thus, the seepage and level changes in groundwater in the confined are not affected.

### 4.3. Settlement of Tunnel

In the graded dewatering stage, the ground settlement along the tunnel is approximately 57.0% of that in open space because of the constraint effect of the tunnel structure. When buildings are located around the tunnel, the ground settlement will be reduced by 3.0% again (Figure 6). As the key factor that affects tunnel safety, the uneven settlement between JS–1 and JS–5 measuring points is 1.7, 0.9, 0.9, and 2.1 mm, respectively. The maximum uneven settlement appears in the range of 45.0–90.0 m away from the central point of the tunnel (JS–3, perpendicular point to deep excavation), which is mainly affected by the frictional resistance of the diaphragm wall.

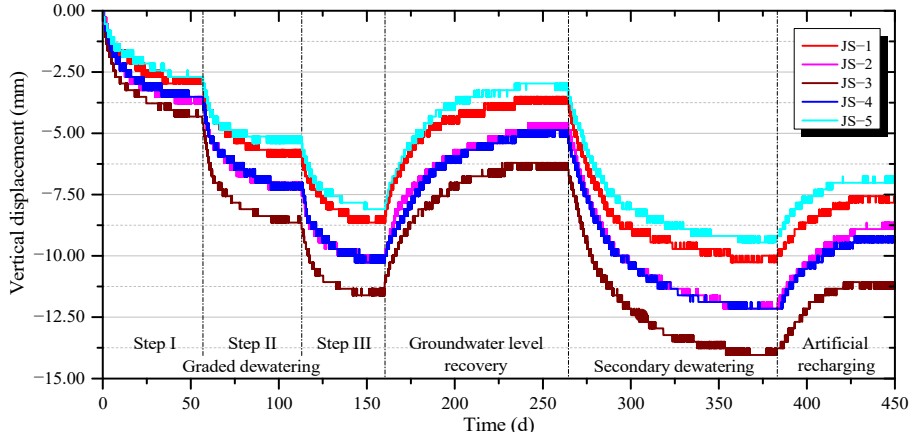

**Figure 6.** Vertical settlement along the tunnel.

The excavation will cause the displacement of walls and the reduction in earth pressure behind the walls, while to separately analyze the land settlement caused by dewatering, the PMMA tube is used to simulate the diaphragm wall, so the structural deformation caused by excavation is ignored. The value of earth pressure changes on the outer wall of the tunnel near the deep excavation is evidently greater than that under the tunnel (Figure 7), thus indicating that the tunnel is subjected to eccentric force during the vertical settlement process, thereby causing its horizontal movement and rotating to the side of the deep excavation [33]. In soft clay, more than 75% of the vertical ground movement may have occurred into the upper annulus of the oval-shaped gap due to the plastic flow of the yielded clay around the tunnel. Based on the oval-shaped gap geometry, the magnitude of the horizontal movement at the tunnel spring line is approximately half of the vertical movement at the tunnel crown [34]. Unfortunately, this phenomenon cannot be accurately measured because of the model size and the measurement methods [35]. When the groundwater level is recovered, the final settlement of each measuring point rebounds to 39.0–55.0% of the previous stage, and the uneven settlement between the JS–1 and JS–5 measuring points is 1.1, 1.5, 1.2, and 1.8 mm, respectively. The maximum uneven settlement appears around the deep excavation and the buildings, which is also influenced by the frictional resistance of the diaphragm wall and building foundations. After groundwater-level dewatering again, the final settlement increases by 13.0–20.0% compared with the first stage, and the maximum uneven settlement that appeared at the same location reached 2.1–3.0 mm.

To control the tunnel's settlement, artificial groundwater recharging is added, as shown in Figure 8 groundwater level drawdown in the aquifer decreased to 13.0 m (recovery 35.8% at −60.0 m under the roof of layer ⑨), 6.0 m (recovery 60.0% at −7.5 m under the roof of layer ⑨), and 11.0 m (recovery 13.7% at −7.5 m under the roof of layer ⑦). The final settlement of each measuring point in this stage is approximately 7.0–10.0 mm, which is rebounded to 70.0–80.0% of the previous stage, and the maximum uneven settlement is 2.4 mm near the buildings. Artificial groundwater recharging can protect the

settlement of the tunnel and the surrounding strata. However, meeting the settlement control requirements is difficult because of its scope and amount.

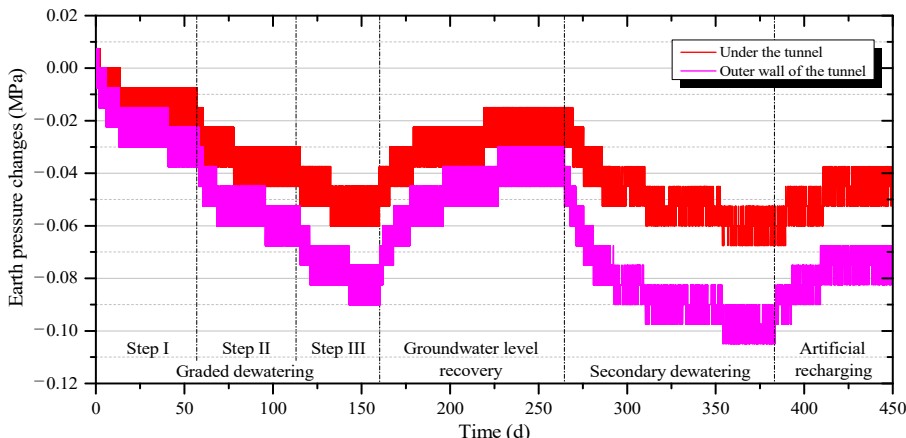

**Figure 7.** Earth pressure around the tunnel at JS–3.

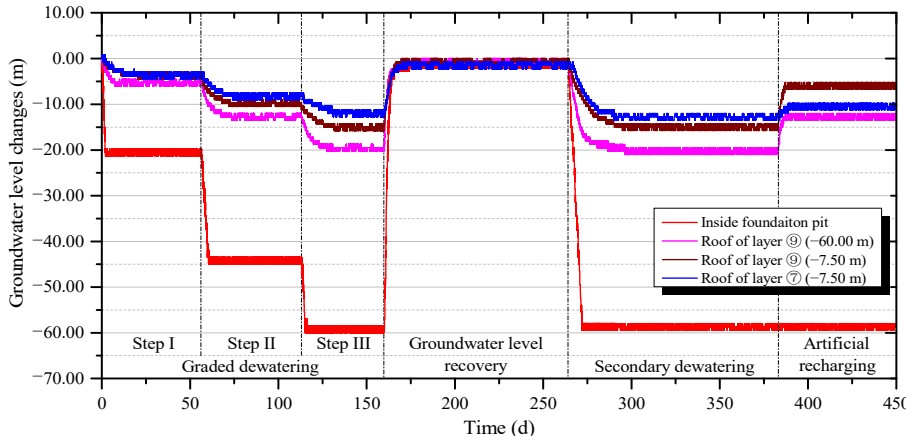

**Figure 8.** Groundwater level changes under the tunnel at JS–3.

## 5. Discussion

When pore water pressure decreases rapidly in the confined aquifer and if the overlying aquitard has excellent water barrier performance and almost no groundwater seepage occurs at the beginning, only vertical settlement and internal expansion will occur in the overlying strata, which are called "coordinated deformation" or "reverse rebound", under the constraints of its own structural stiffness and surrounding environment. Pore water pressure decreases first in the confined aquifer. Then, a hydraulic gradient occurs between the top of the confined aquifer and the bottom of the overlying aquitard. When the overlying aquitard reaches a certain degree of hydraulic permeability, vertical groundwater overflow will be driven by the hydraulic gradient from the underlying confined aquifer. With the increase in time, a hydraulic gradient forms in the aquitard and increases gradually, and its rising range is limited by the thickness and permeability of the overlying strata and the duration of dewatering. The soil shows compression under the drive of seepage within the range of hydraulic gradient. Meanwhile, the soil above it is still affected by the coordinated deformation and shows expansion (Figure 9).

As shown in Figure 10a, when buildings are located around the deep excavation, the pile foundation of buildings settles along with the soil of the bearing layer, and the deformation of the strata within the pile length is limited under the influence of its frictional resistance. In the initial stage of deep excavation dewatering, pile foundations settle together with the confined aquifer. Furthermore, the soil around buildings shows larger expansion because of the overlying strata's coordinated deformation and the piles' frictional

resistance. With the extension of time, groundwater overflow and compression gradually develop in the overlying aquitard, and the compression deformation of the soil is restricted by the frictional resistance of the piles again. Compared with open spaces, the final ground settlement around buildings is larger, and the rebound after dewatering is smaller. As for buildings, the uneven settlement of the bearing layer in the horizontal position leads to its tilt deformation (Figure 10b), and the friction resistance in different directions during two stages of deformation will produce complex additional stress on the piles and may cause damage. In the vertical position, the maximum uneven settlement between the foundation bearing layer and the ground reaches 4.95 mm at JM–2. This maximum level may cause fractures between the main structure of the buildings and the infrastructure or pipelines in their surroundings, which mostly occur in the surrounding parts of building groups.

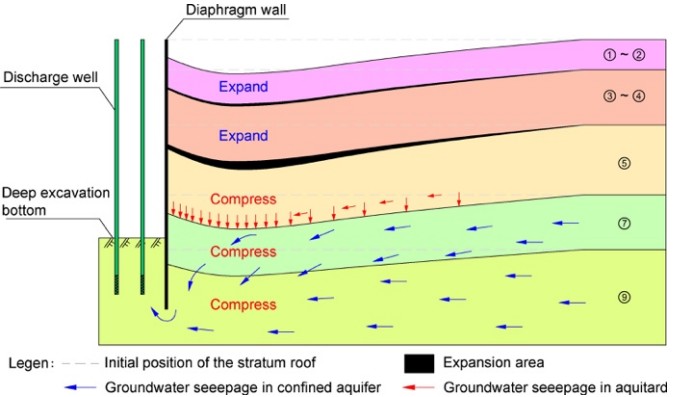

**Figure 9.** Conceptual schema of stratified settlement in strata.

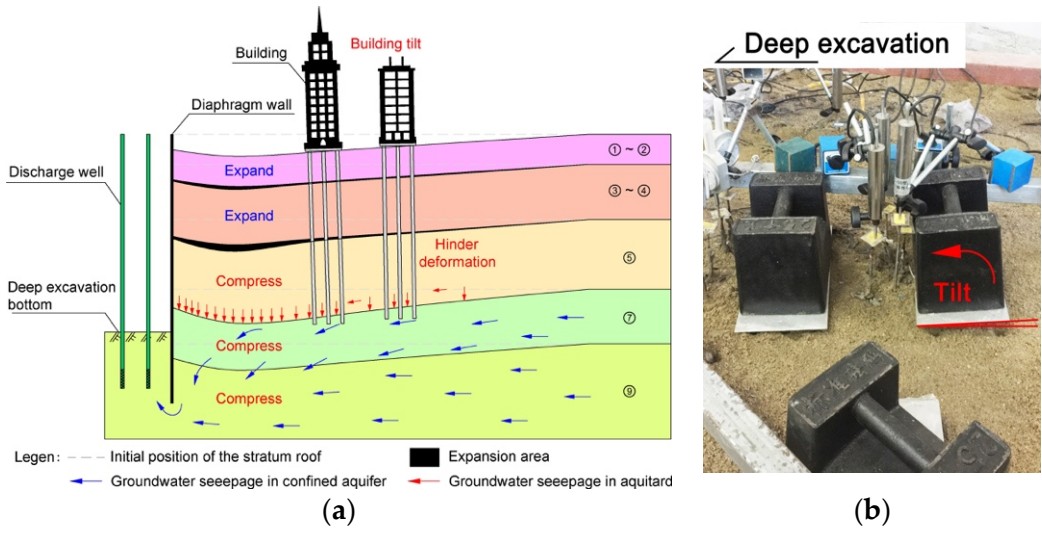

(**a**)          (**b**)

**Figure 10.** Stratified settlement in strata around buildings: (**a**) Conceptual schema; (**b**) Buildings' tilt in physical model.

When a tunnel is located around, its structural integrity will restrict the deformation of the surrounding soil and itself. The ground settlement above it decreases to a certain range, and the earth pressure on the bottom and outer walls of the tunnel is also decreased. Given the uneven earth pressure around the tunnel, horizontal and torsional displacement may occur in the process of vertical displacement along with the soil (Figure 11), thereby threatening the sealing performance of the tunnel.

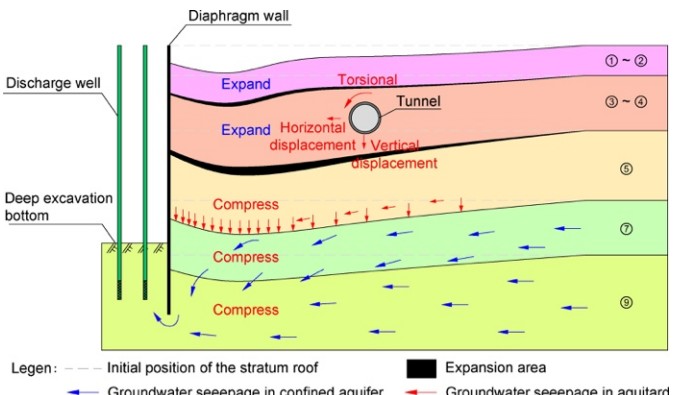

**Figure 11.** Conceptual schema of stratified settlement in strata around the tunnel.

## 6. Conclusions

In this study, a physical model test was conducted to examine the law of stratified settlement in MAMA. A similar material was developed, which achieved deformation and seepage similar to real conditions. Buildings and a tunnel were added inside the physical model to analyze the impact of environmental conditions on the stratified settlement in MAMA. The following conclusions were obtained:

- According to whether the overlying aquitard is coupled with groundwater seepage, the process of dewatering-induced stratified settlement in MAMA can be divided as compression in the confined aquifer, coordinated deformation in the overlying strata, and seepage and compression in the overlying strata. The soil shows compression under the drive of seepage within the range of the hydraulic gradient, whereas the soil above it is still affected by the coordinated deformation and shows expansion.

- Stratified settlement around the diaphragm wall is confined and expanded at the top strata (layer ③ + ④ in this test). The maximum ground settlement occurs approximately 45.0 m away from the deep excavation. When the groundwater level is recovered, the deformation of the sandy strata (layers ⑦ and ⑨) rebounds quickly. Meanwhile, such a rebound is difficult in the clayey strata (layers ③ + ④ and ⑤) due to plastic strain, and about 50.0% of the ground settlement is rebound. The large-scale and uninterrupted excavation and dewatering of the underground space are the main reasons for the continuous development of land subsidence.

- The existence of artificial underground structures will limit the deformation and internal expansion of the strata. Affected by the frictional resistance of buildings' pile foundations, the settlement of each stratum within its length range is decreased, and the ground settlement is approximately 80.0% of that in open space. Therefore, dewatering-induced uneven settlement will cause damage to existing foundations and underground structures.

- When a tunnel exists, the ground settlement above it is decreased by about 45.0% because of its structural integrity. The maximum uneven settlement appears in the range of 45.0–90.0 m away from the perpendicular point to the deep excavation, especially when buildings exist around it. Artificial groundwater recharging can reduce the settlement of the tunnel and the surrounding strata. However, meeting the settlement control requirements is difficult because of their scope and amount.

**Author Contributions:** X.L. and J.W. carried out the main research task and wrote the manuscript. X.L. proposed the original idea and contributed to the revision of the obtained results and the whole manuscript. X.L., N.X. and Y.L. performed the physical model test. T.Y., L.W. and X.H. performed data analysis. All authors have read and agreed to the published version of the manuscript.

**Funding:** This work is supported by the National Natural Science Foundation of China (grant No. 41907230), Science and Technology Commission of Shanghai Municipality (grant Nos. 18DZ1201301, 19DZ1200901), Open Fund of Key Laboratory of Land Subsidence Monitoring and Prevention, Ministry of Natural Resources of China (grant No. KLLSMP202101).

**Institutional Review Board Statement:** Not applicable.

**Informed Consent Statement:** Not applicable.

**Data Availability Statement:** The data presented in this study are available on request from the corresponding author.

**Conflicts of Interest:** The authors declare no conflict of interest.

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
