# Peer review of "Dewatering-Induced Stratified Settlement around Deep Excavation: Physical Model Study"

_applsci, doi:10.3390/app12188929_

Round 1
Reviewer 1 Report
In this paper, a physical model test was taken out to study the law of stratified settlement in MAMA, and similar material was developed and achieved deformation and seepage similar to real conditions. Buildings and tunnels were added inside the physical model to analyze the impact of environmental conditions on stratified settlements in MAMA. In general, it is well organized. However, there are some issues to be addressed before publication:
Professional English proofreading and editing are required - there are many misuses of words and grammar mistakes in the manuscript.
Line 218: what's the size of the sensors? Did the sensor affect the movement of the soils?
Line 242: why the duration time under each groundwater level drawdown is set as 90h? Did you consider the scale effects of time? The same question applies to other times selected in the paper.
Line 257: Did the 180-day consolidation cause water content change in the soil like evaporation?
Could the authors compare the data in the literature with your own results in the paper?
Author Response
Dear Editors and Reviewers,
Thank you very much for your detailed comments and many efforts to help us improve our manuscript (applsci-1869370) entitled “Dewatering Induced Stratified Settlement Around Deep Excavation: Physical Model Study”. Based on the comments, we have made careful revisions on the original manuscript. All revisions made to the text are marked in red so that they may be easily identified. Once again, we acknowledge your comments and constructive suggestions very much, which are valuable in improving the quality of our manuscript.
Here below is our respond to the reviewer’s comments.
Reviewer #1:
Comment: Professional English proofreading and editing are required - there are many misuses of words and grammar mistakes in the manuscript.
Response: It has been revised. We have revised and improved the language under the help of a professional language agency.
Comment: Line 218: what's the size of the sensors? Did the sensor affect the movement of the soils?
Response: To avoid the influence of the sensor on soil deformation, small-sized sensors were selected. The pore water pressure sensor is φ 1.0 cm and 2.0 cm in thickness, and the earth pressure sensor is φ 2.5 cm and 0.5 cm in thickness. The settlement pole is made of a φ 0.2 cm pole and two 1.5 cm * 1.5 cm plates at both ends and a φ0.3 cm sleeve was placed on the outside of the 0.2cm pole, so it can move freely in the vertical. So, because of the use of small size sensors and vertical free movement of the settlement pole, the soil deformation was not affected.
It has been revised. See lines 223-227.
Comment: Line 242: why the duration time under each groundwater level drawdown is set as 90h? Did you consider the scale effects of time? The same question applies to other times selected in the paper.
Response: Because the time similarity ratio is set as , 90 hours in the model equals 56 days in the prototype, the duration of groundwater level drawdown is determined by the actual deep excavation process. In the results, all the time has been converted to the prototype state by the similar ratio.
It has been revised. See lines 248.
Comment: Line 257: Did the 180-day consolidation cause water content change in the soil like evaporation?
Response: Two circulating water supply systems are set outside to control the groundwater levels of two variable hydraulic boundaries separately. In the consolidation stage, the hydraulic boundaries were always maintained groundwater recharge, and soil water content didn’t change.
Comment: Could the authors compare the data in the literature with your own results in the paper?
Response: Thank you for your suggestion, but this physical model test is based on a simplified geological model and hypothetical excavation conditions, it cannot be directly compared with the existing literature data. This work will be the focus of our next research.

Reviewer 2 Report
Excellent paper with well organized structure and descriptions.
Few suggestions:
1) at #4.3 comment the possible influence of excavation technique on the displacement of walls and earth pressure control of soil conditioning
2) References
a) For the discussion and procedure see the Open Access paper
Tunnel static behavior assessed by a probabilistic approach to the back-analysis
By Oreste, P. et al
American Journal of Applied Sciences, 2012, 9(7), pp. 1137–1144
doi 10.3844/ajassp.2012.1137.1144
b) for the discussion on settlement interpretation and comparison between model and theory:
Loganathan N, Poulos HG (1998)
Analytical prediction for
tunneling-induced ground movements in clays.
J Geotech Geoenviron Eng 124(9):846–856
Regards
Author Response
Dear Editors and Reviewers,
Thank you very much for your detailed comments and many efforts to help us improve our manuscript (applsci-1869370) entitled “Dewatering Induced Stratified Settlement Around Deep Excavation: Physical Model Study”. Based on the comments, we have made careful revisions on the original manuscript. All revisions made to the text are marked in red so that they may be easily identified. Once again, we acknowledge your comments and constructive suggestions very much, which are valuable in improving the quality of our manuscript.
Here below is our respond to the reviewer’s comments.
Reviewer #2:
Comment: 1) at #4.3 comment the possible influence of excavation technique on the displacement of walls and earth pressure control of soil conditioning
Response: The excavation of foundation pit will cause the displacement of walls and the reduction of earth pressure behind the walls, while in order to separately analyze the land settlement caused by dewatering, the PMMA tube is used to simulate the diaphragm wall, thus the structural deformation caused by excavation is ignored. The tunnel is an existing structure, it is used for the deformation characteristic analyze caused by foundation pit dewatering.
It has been revised. See lines 351-354 and 358-362.
Comment: 2) References
a) For the discussion and procedure see the Open Access paper:
Oreste, P. et al. Tunnel static behavior assessed by a probabilistic approach to the back-analysis. American Journal of Applied Sciences, 2012, 9(7), pp. 1137–1144. doi: 10.3844/ajassp.2012.1137.1144
b) for the discussion on settlement interpretation and comparison between model and theory:
Loganathan N, Poulos HG (1998) Analytical prediction for tunneling-induced ground movements in clays. J Geotech Geoenviron Eng 124(9):846–856
Response: The references have been cited and explained accordingly. See line 362, 364, 508-509, and 517-518.
